# Safety and Efficacy of Nirsevimab in a Universal Prevention Program of Respiratory Syncytial Virus Bronchiolitis in Newborns and Infants in the First Year of Life in the Valle d’Aosta Region, Italy, in the 2023–2024 Epidemic Season

**DOI:** 10.3390/vaccines12050549

**Published:** 2024-05-17

**Authors:** Alessandra Consolati, Mariapaola Farinelli, Paolo Serravalle, Christine Rollandin, Laura Apprato, Susanna Esposito, Salvatore Bongiorno

**Affiliations:** 1Hygiene and Public Health Structure, Prevention Department, Local Health Authority Valle d’Aosta, 11100 Aosta, Italy; aconsolati@ausl.vda.it (A.C.); mfarinelli@ausl.vda.it (M.F.); lapprato@ausl.vda.it (L.A.); sbongiorno@ausl.vda.it (S.B.); 2Paediatrics and Neonatology Complex Structure, Mother and Child Department, Local Health Authority Valle d’Aosta, 11100 Aosta, Italy; pserravalle@ausl.vda.it; 3School of Specialization in Health Statistics and Biometrics, University of Turin, 10124 Turin, Italy; crollandin@ausl.vda.it; 4Simple Departmental Structure Preventive Medicine and Risk Management, Strategic Management of the Local Health Authority of Valle d’Aosta, 11100 Aosta, Italy; 5Pediatric Clinic, Department of Medicine and Surgery, University of Parma, 43126 Parma, Italy

**Keywords:** bronchiolitis, monoclonal antibody, nirsevimab, respiratory syncytial virus, RSV

## Abstract

Respiratory syncytial virus (RSV) bronchiolitis remains a significant global health burden, particularly in newborns and infants during their first year of life. The quest for an effective preventive strategy against RSV has long been sought, and recent developments have shown promise in the form of nirsevimab, a monoclonal antibody specifically designed for RSV prophylaxis. Valle d’Aosta was the first Italian region to propose universal prophylaxis with nirsevimab for newborns and infants in their first epidemic season as early as 2023–2024. This study describes the effectiveness and safety of the universal prevention program of RSV bronchiolitis using the monoclonal antibody nirsevimab in children resident in Valle d’Aosta born during the 2023–2024 epidemic season. There were 556 neonates born from 1 May 2023 to 15 February 2024. The risk of hospitalization for RSV bronchiolitis in 2023–2024 was 3.2%, compared to 7% in the 2022–2023 epidemic season (*p* < 0.001). After the start of the prophylaxis campaign with nirsevimab, the risk of hospitalization was 8.3% in the sample of infants who did not adhere to the prophylaxis, while no child in the sample of those treated (*p* < 0.001) was hospitalized for bronchiolitis. Few mild transient side effects were reported. This study shows the efficacy and safety of universal prophylaxis with nirsevimab in neonates, making Valle d’Aosta the first Italian region to offer universal prophylaxis to newborns without risk factors for RSV complications. Future research could further explore its long-term impact and cost-effectiveness.

## 1. Introduction

Respiratory syncytial virus (RSV) bronchiolitis poses a substantial global health burden, especially among newborns and infants within their inaugural year of life [1,2,3,4,5]. As the predominant cause of bronchiolitis and pneumonia in infants on a global scale, RSV infections often reach a peak during the winter months [6,7]. Furthermore, the repercussions of RSV extend beyond the acute illness, with potential long-term respiratory complications such as recurrent wheezing and asthma [8,9,10,11]. Consequently, there is a pressing demand for the implementation of robust preventive strategies to mitigate the impact of RSV on infant health. Given its prevalence and the potential for persistent respiratory issues, addressing RSV bronchiolitis effectively is imperative for safeguarding the well-being of infants worldwide. Efforts to develop and deploy effective preventive measures, alongside ongoing research to better understand RSV pathogenesis and transmission dynamics, are crucial steps in alleviating the burden of RSV-related illnesses in infants and improving their long-term respiratory outcomes.

Despite advancements in medical care, RSV-related morbidity and mortality persist, leading to substantial economic costs and emotional strain on families [12]. Currently, the primary approach for preventing RSV infections involves the utilization of the monoclonal antibody palivizumab. However, the administration of palivizumab is reserved for specific populations deemed at high risk of complications associated with RSV infection. These include preterm infants born prematurely with a gestational age less than 29 weeks, infants diagnosed with hemodynamically significant heart disease, and individuals with pulmonary abnormalities or neuromuscular diseases that hinder the effective clearance of secretions from the upper airways, as well as infants with primary or secondary immunodeficiencies [13,14]. The usage of palivizumab is constrained by several factors, including its restricted therapeutic indication, which limits its applicability to specific patient populations. Additionally, its high cost presents a significant barrier to widespread adoption, making it less accessible to healthcare systems and patients alike. Moreover, due to its relatively short half-life, palivizumab necessitates frequent administrations, typically on a monthly basis, particularly during peak epidemic periods when the risk of respiratory syncytial virus (RSV) infection is heightened. This requirement for regular dosing adds to the logistical challenges and resource burdens faced by healthcare providers and patients. As a result, despite its efficacy in preventing severe RSV-related illnesses in certain high-risk populations, the practical limitations associated with palivizumab use underscore the need for alternative prevention strategies and interventions. More than 60% of children acquire respiratory syncytial virus (RSV) by the age of one, and nearly all experience at least one infection by age two. However, the subset of children eligible for palivizumab prophylaxis is relatively small, constituting only 4–6% of the pediatric population [13,15]. Between 70% and 90% of hospitalized newborns due to RSV infection are born at term and lack underlying chronic pathologies, rendering them ineligible for palivizumab prophylaxis [16,17,18,19].

The pursuit of an effective preventive strategy against RSV has been ongoing for years [20], and recent breakthroughs have illuminated the potential of nirsevimab, a monoclonal antibody tailored for RSV prophylaxis [21,22,23]. Being a humanized monoclonal antibody, nirsevimab offers a novel approach to RSV prevention by specifically targeting the fusion (F) protein, a critical component of the virus essential for its infectivity. What sets nirsevimab apart from previous prophylactic options is its extended half-life and potent neutralizing activity against a wide spectrum of RSV strains, including those resistant to other monoclonal antibodies [24]. These distinctive attributes, such as providing protection for at least 5 months with just a single administration during the epidemic season, render nirsevimab an appealing candidate for universal RSV prophylaxis in newborns and infants. Clinical trials evaluating the safety and efficacy of nirsevimab have yielded promising results [25,26]. Notably, a large phase 2b trial demonstrated a significant reduction in medically attended RSV lower respiratory tract infections (LRTIs) and RSV-related hospitalizations compared to placebo, with a favorable safety profile [25]. Subsequent phase 3 trials further validated these findings, firmly establishing nirsevimab as a highly effective prophylactic agent against severe RSV disease [26]. Besides its efficacy, the safety profile of nirsevimab holds significant importance, especially for the vulnerable populations of newborns and young infants. Clinical trials have consistently reported a low incidence of adverse events post-nirsevimab administration, with no evidence indicating an increased risk of serious adverse events compared to placebo [25,26]. This favorable safety profile underscores the potential of nirsevimab as a well-tolerated preventive intervention for RSV bronchiolitis. Following the marketing authorization for nirsevimab in the European Union countries [27], several nations, including France, Spain (Galicia), Ireland, and Luxembourg, have incorporated universal prophylaxis with nirsevimab into their national prevention plans. This proactive step highlights the recognition of nirsevimab’s efficacy and safety profile and signifies a significant advancement in the prevention of RSV infections among newborns and infants on a broader scale. As more countries embrace nirsevimab as a cornerstone of their RSV prevention strategies, there is optimism for reduced disease burden and improved health outcomes in pediatric populations worldwide [28,29,30].

In the Valle d’Aosta Region, like in other parts of Italy, recent epidemic seasons have witnessed a notable surge in cases of RSV infections, coupled with an escalation in their severity. Throughout the 2022–2023 epidemic season, the hospitalization risk for RSV among infants experiencing their inaugural epidemic season stood at 7%, surpassing the national average of 4% [31]. This translated to 51 hospitalizations out of 729 new births, underscoring the significant healthcare burden imposed by RSV-related admissions. The demands on healthcare resources were substantial, owing to both the prolonged duration of hospital stays and the heightened intensity of care necessitated by the severity of RSV infections. Valle d’Aosta emerged as a trailblazer in Italy by pioneering the concept of universal prophylaxis with nirsevimab for newborns and infants during their first epidemic season as early as 2023–2024. With a population of approximately 123,000 residents and 702 new births recorded in 2023, Valle d’Aosta ranks as the smallest region in the country with the lowest population density. This unique demographic composition, characterized by a single Local Health Unit and a solitary hospital birth center, facilitated the seamless implementation of the passive immunization initiative utilizing nirsevimab. By spearheading universal prophylaxis with nirsevimab, Valle d’Aosta aimed to mitigate the incidence and severity of RSV infections among vulnerable newborns and infants, thereby alleviating the strain on healthcare resources and improving patient outcomes. This proactive approach underscores the region’s commitment to safeguarding the health and well-being of its youngest residents, setting a precedent for other regions across Italy to emulate. This study describes the effectiveness and safety of the universal prevention program for RSV bronchiolitis using the monoclonal antibody nirsevimab in children resident in Valle d’Aosta born during the 2023–2024 epidemic season.

## 2. Methods

This is a prospective observational cohort study relating to the incidence of hospitalization for RSV bronchiolitis or pneumonia in two populations of children with similar demographic and comorbidity characteristics, either subjected to or not subjected to nirsevimab prophylaxis. The secondary end-point of the study was the evaluation of the safety of nirsevimab by monitoring the onset of short-term adverse effects. All individuals born between 1 May 2023 and 15 February 2024 and residing in Valle d’Aosta were considered for inclusion in the study. However, those with pre-existing risk factors that had already undergone palivizumab prophylaxis were excluded. These risk factors included preterm infants born prematurely with a gestational age of less than 29 weeks, infants diagnosed with hemodynamically significant heart disease, and individuals with pulmonary abnormalities or neuromuscular conditions impairing the ability to clear secretions from the upper airways, as well as infants with primary or secondary immunodeficiencies. Identification of newborns and infants eligible for nirsevimab prophylaxis was conducted through the utilization of Local Health Unit information systems and the assisted registry. This process ensured that candidates meeting the criteria for nirsevimab prophylaxis were accurately identified and included in the immunization campaign. By leveraging existing healthcare infrastructure and information systems, healthcare providers could efficiently identify and target those individuals who would benefit most from nirsevimab prophylaxis, thus optimizing the allocation of resources and maximizing the impact of the intervention in preventing RSV infections in the pediatric population of Valle d’Aosta.

Leading up to the commencement of the immunization campaign, extensive training sessions were conducted from 1 September 2023 to 30 November 2023. These sessions involved all personnel engaged in the initiative, including hospital and primary care pediatricians, public health doctors, pediatric nurses, midwives, health assistants, and administrative staff. The primary objective of these training endeavors was to enhance adherence to the program among families by fostering collaboration among child health professionals across various sectors, including hospital, local, and prevention areas. The dissemination of the project’s objectives, guidelines, and protocols was carried out through both internal and external communication channels. These efforts included engaging with press bodies and utilizing internal communication platforms within the Valle d’Aosta Region Local Health Unit. By disseminating comprehensive information and facilitating effective communication channels [32], the aim was to ensure that all stakeholders were well informed and aligned with the objectives of the immunization campaign. Through these coordinated efforts, the campaign sought to maximize participation and optimize the delivery of nirsevimab prophylaxis to eligible newborns and infants within the Valle d’Aosta region.

Nirsevimab became accessible starting from 20 December, procured through direct importation from the manufacturer in France. This acquisition process adhered to the procedure established by the Italian Ministry of Health for drugs not yet available on the Italian market. To ensure widespread dissemination and accessibility, the Public Health and Hygiene Service (SISP) reached out to parents of neonates born between 1 May and 18 December 2023, as well as those born outside the region. Correspondence was facilitated via letters dispatched through ordinary postal services, containing comprehensive information about the prophylaxis, along with consent forms and instructions for scheduling appointments.

The actual administration of nirsevimab took place leveraging the structural and human resources of the SISP, centrally located at the Aosta headquarters. For infants discharged from the hospital nursery between 20 December 2023 and 15 February 2024, prophylaxis was provided directly within the Neonatology Unit on the day of discharge. Throughout the entire process, utmost attention was given to obtaining written informed consent from all parents prior to administering the drug. This ensured compliance with ethical and regulatory standards while also respecting parental autonomy in decision-making regarding their child’s healthcare. Additionally, consent encompassed authorization for the processing of personal data, guaranteeing compliance with data protection regulations and safeguarding individuals’ privacy rights throughout the prophylaxis administration process.

To monitor the potential onset of side effects, telephone interviews were conducted at 7 and 14 days post-administration of nirsevimab. These interviews served as a proactive measure to assess and document any adverse reactions following prophylaxis administration. Additionally, data pertaining to hospitalizations for RSV bronchiolitis were obtained from the Local Health Unit information systems. This allowed for comprehensive tracking and analysis of hospitalization trends related to RSV infection within the study population.

Data analysis was conducted utilizing STATA^®^ Software (Release 12, College Station, TX, USA). Descriptive statistics, including absolute frequency and percentage, were employed for the analysis of dichotomous and categorical variables. Comparisons between such variables were carried out using appropriate statistical tests, including the chi-square test for frequencies exceeding 5 and Fisher’s exact test for frequencies below or equal to 5. Furthermore, Microsoft Excel^®^ was utilized for graphical representation of the data, facilitating visual interpretation and analysis of key findings. This comprehensive approach to data analysis enabled a thorough evaluation of nirsevimab prophylaxis efficacy and safety, providing valuable insights into its potential impact on RSV-related hospitalizations and adverse events within the studied population.

## 3. Results

Table 1 provides an overview of the adherence rate to nirsevimab prophylaxis within the enrolled population. A total of 556 neonates were born between 1 May 2023 and 15 February 2024. Among the 461 neonates born between 1 May 2023 and 18 December 2023, 13 infants who received palivizumab were excluded from the analysis. Out of the remaining 448 candidates, 292 (65%) adhered to nirsevimab prophylaxis. For the 95 neonates born between 19 December 2023 and 15 February 2024, three were excluded due to non-residency in the region and three underwent palivizumab prophylaxis. Subjects with prematurity or other comorbidities received prophylaxis with palivizumab as indicated, according to the current guidelines of the Italian Society of Neonatology for the 2023–2024 epidemic season, and also considering the late (20 December) availability of nirsevimab. Among the remaining 89 newborns, 77 (86%; *p* < 0.001 compared to the previous period) adhered to the prophylaxis, receiving the drug before discharge from the Neonatology Unit. Overall, the participation rate was 69%, indicating a substantial level of adherence to nirsevimab prophylaxis among eligible neonates within the study population. This high level of adherence suggests robust acceptance and implementation of the prophylactic regimen, potentially contributing to the effective prevention of RSV infections among infants in the Valle d’Aosta region.

During the current epidemic season, until 15 February 2024, there were a total of 29 hospitalizations for RSV bronchiolitis in the Pediatrics department of the Regional Hospital. Among these, 18 admissions were recorded for children born after 1 May 2023. By comparison, in the previous epidemic season, by the same date, there were 61 hospitalizations, with 47 involving children born after 1 May 2022. The periodic risk of hospitalization for RSV from the onset of the epidemic season until 15 February 2024 in the cohort under study was 3.2%, contrasting with the 7% prevalence observed in the 2022–2023 epidemic season. Following the initiation of the prophylaxis campaign with nirsevimab, the risk of hospitalization for RSV bronchiolitis among children who did not adhere to the prophylaxis was 8.3% (14 out of 168). Conversely, none of the children who received treatment (0 out of 369) were hospitalized for bronchiolitis, yielding a statistically significant difference (*p* < 0.001). Figure 1 illustrates the trend of hospitalizations for RSV bronchiolitis in the Aosta Valley during the last two epidemic seasons, demonstrating a notable reduction in hospitalizations following the implementation of the nirsevimab prophylaxis campaign.

Side effects observed within two weeks post-nirsevimab administration typically manifested within 48 h post-treatment. These effects were generally mild and short-lived, lasting 1–2 days, with consistent outcomes across both administration periods. The reported side effects included fever (6.5%), local reactions at the injection site (4%), and barely consolable crying (0.4%). Notably, none of the reported cases necessitated additional medical visits, and no instances of major adverse effects were documented. Figure 2 provides a visual representation of the occurrence and distribution of these side effects, highlighting their generally benign nature and limited impact on the recipients of nirsevimab prophylaxis.

## 4. Discussion

This study heralds a significant milestone, showcasing the effectiveness and safety of universal prophylaxis with nirsevimab in neonates, thereby positioning Valle d’Aosta as the pioneering Italian region to extend universal prophylaxis to newborns without risk factors for RSV complications.

The results of this study underscore a marked reduction in the risk of RSV-related hospitalization among neonates who received nirsevimab compared to those who did not. Notably, none of the children immunized with nirsevimab required hospitalization for RSV bronchiolitis, indicating the robust efficacy of nirsevimab in preventing RSV infections within this vulnerable population. In the absence of specific and effective pharmacological therapies targeting RSV [33,34], a strategy centered on universal prevention emerges as a compelling approach, ensuring broad coverage across the population and equitable access, thereby significantly enhancing the protection of newborns’ and infants’ health. In Valle d’Aosta, hospitalizations for RSV infection in children not in their first epidemic season (i.e., born before 1 May 2023 and, therefore, not candidates for prophylaxis with nirsevimab) in the 2022–2023 epidemic season were 13, and in the epidemic season 2023–2024 were 11. This highlights that the RSV epidemic in the Valle d’Aosta region was similar in the two seasons and supports that the decrease of the RSV-related hospitalization was due to the effect of the universal nirsevimab program but not the lower epidemic rate. On the other hand, Italian data in the 2022–2023 and 2023–2024 epidemic seasons confirm high RSV circulation, with no differences between the two periods [35].

Moreover, from an economic sustainability standpoint, the impact assessment must encompass direct savings realized through reduced hospital admissions, cessation of selective prophylaxis with palivizumab, and indirect savings stemming from diminished social care costs borne by parents for children afflicted by RSV infection (e.g., medical expenses and lost work days) or enduring long-term consequences of RSV infection (e.g., recurrent wheezing or asthma) [36]. Nirsevimab’s extended dosing interval (once every RSV season) and its potential to provide protection throughout the RSV season could translate into substantial cost savings over time. Additionally, its single-dose administration further simplifies logistics and reduces healthcare resource utilization. While the upfront cost of universal RSV prevention may be high, the potential long-term savings and improved clinical outcomes associated with nirsevimab treatment could make it a cost-effective option for RSV management. Although further economic evaluations are needed to comprehensively assess the cost-effectiveness of nirsevimab compared to standard therapy, data underscore the value proposition of universal prophylaxis with nirsevimab in mitigating the burden of RSV-related illnesses.

Interestingly, our findings align closely with those reported in Spain between October 2023 and January 2024, where a universal immunization program featuring nirsevimab achieved coverage rates ranging from 79% to 99% [37]. This program demonstrated a minimum efficacy of 70% in preventing hospitalizations among infants with lower respiratory tract infections (LRTIs) positive for RSV, albeit without conferring protection against RSV-negative LRTI-related hospitalizations [37]. Such parallels underscore the consistency and reproducibility of the beneficial outcomes associated with universal prophylaxis with nirsevimab across different geographical settings. These results not only affirm the efficacy and safety of universal prophylaxis with nirsevimab in neonates but also highlight its potential to deliver substantial economic benefits and align with international experiences, positioning it as a cornerstone in the global fight against RSV-related morbidity and mortality. Through concerted efforts to implement and optimize universal prophylaxis programs, we stand poised to achieve significant strides in safeguarding the health and well-being of newborns and infants worldwide.

In Luxembourg, the year 2023 marked a significant milestone for the introduction of nirsevimab immunization against RSV, achieving an estimated coverage rate of 84% among newborns [38]. This proactive initiative yielded promising outcomes, notably a decline in pediatric RSV-related hospitalizations, particularly among infants aged under 6 months, compared to the corresponding period in 2022. Noteworthy shifts were observed in the demographics of hospitalized children during this period, with the mean age increasing from 7.8 months in 2022 to 14.4 months in 2023 (*p* < 0.001). Concurrently, there was a notable reduction in the duration of hospital stays, with a decrease from 5.1 days in 2022 to 3.2 days in 2023 (*p* < 0.001). Furthermore, among infants under 6 months old, admissions to the intensive care unit appeared to decrease substantially, dropping from 28 cases in the previous period to 9 cases in 2023 [38]. Meanwhile, in the United States, nirsevimab demonstrated remarkable effectiveness, exhibiting a 90% efficacy (95% confidence interval [CI] = 75−96%) against RSV-associated hospitalization [39]. Impressively, recipients experienced a median time from receipt to symptom onset of 45 days, highlighting the rapid onset of protective effects post-administration. Moreover, a recent multicenter study conducted in the United States observed no discernible increase in medically attended LRTIs or evidence of antibody-dependent enhancement of infection or disease severity among nirsevimab recipients compared to placebo recipients [40]. These findings underscore the substantial impact of nirsevimab in mitigating the burden of RSV-related illnesses, both in Luxembourg and the United States. The introduction of nirsevimab immunization has not only led to a reduction in hospitalizations and intensive care admissions but has also demonstrated high efficacy in preventing RSV-associated hospitalization, thereby offering significant benefits to pediatric populations [38,39,40]. The favorable safety profile and rapid onset of protective effects further solidify nirsevimab’s position as a key intervention in the fight against RSV infections, paving the way for improved pediatric health outcomes on a global scale.

Our study delves into the safety profile of nirsevimab, shedding light on its favorable safety profile with the absence of significant adverse events or safety concerns associated with its administration [25,26]. Building upon previous research, which has also highlighted the benign safety profile of nirsevimab, our findings reaffirm its suitability as a prophylactic intervention in neonates.

The success of our study owes much to the collaborative efforts of various healthcare professionals, including neonatologists, hospital pediatricians, primary care pediatricians, and hygienists. This multidisciplinary approach, coupled with the forward-thinking political decision to endorse and fund our innovative proposal, played a pivotal role in promoting and implementing our study. A similar observation was made in Spain, where the adoption of flexible individualized appointments and educational initiatives was deemed crucial for fostering nirsevimab uptake [41].

Despite its strengths, our study is not without limitations. Conducted in a small region with a low population density, it may not fully capture the complexities and nuances of larger, more diverse populations. However, it represents a real-world experience that offers valuable insights into the feasibility and efficacy of universal prophylaxis with nirsevimab. By engaging both the hospital and territorial healthcare sectors, our study underscores the potential of universal nirsevimab prophylaxis in alleviating the burden on healthcare systems and enhancing outcomes for newborns globally. Overall, our findings underscore the importance of collaborative efforts and forward-thinking policies in promoting innovative preventive measures like universal nirsevimab prophylaxis. While our study is limited in scope, its real-world implications highlight the transformative potential of such interventions in improving pediatric healthcare delivery and outcomes. Moving forward, continued research and implementation efforts are essential to maximize the benefits of universal nirsevimab prophylaxis and address the ongoing challenges posed by RSV-related morbidity and mortality in newborns.

Continued surveillance is imperative to monitor for any emerging patterns of resistance or shifts in RSV epidemiology following the widespread adoption of nirsevimab prophylaxis. Ongoing monitoring efforts can help identify any changes in RSV strains or trends in disease burden, informing strategies for RSV prevention and control. Specifically, examining its impact on preterm infants and those with underlying comorbidities could provide tailored recommendations for at-risk populations. By elucidating how nirsevimab performs in different clinical contexts, subgroup analyses can inform targeted interventions to maximize its benefits for those most in need.

## 5. Conclusions

This study offers compelling evidence supporting the universal use of nirsevimab for prophylaxis in neonates, presenting a robust case for its efficacy and safety in mitigating RSV-related bronchiolitis. By demonstrating the effectiveness of nirsevimab in preventing RSV infections among newborns, this research lays a solid foundation for its potential to revolutionize the approach to RSV prevention in this vulnerable population. While the findings of this study provide promising insights into the benefits of nirsevimab prophylaxis, further research is warranted to explore its long-term impact and cost-effectiveness. Longitudinal studies could investigate the sustained efficacy of nirsevimab over extended periods, shedding light on its durability and potential for lasting protection against RSV infection. Moreover, economic evaluations could assess the overall cost-effectiveness of implementing universal nirsevimab prophylaxis compared to alternative preventive strategies, considering factors such as healthcare resource utilization and societal costs. Furthermore, subgroup analyses stratifying the population on the basis of risk factors for serious RSV disease (such as prematurity, twins, gender, breastfeeding, or parental smoking) could yield valuable insights into the differential effectiveness of nirsevimab across various risk strata. While this study provides compelling evidence in support of universal nirsevimab prophylaxis for neonates, future research endeavors should delve deeper into its long-term efficacy, cost-effectiveness, and impact across diverse patient populations. By addressing these key research questions, we can advance our understanding of nirsevimab’s role in preventing RSV infections and optimize its implementation to safeguard the health of newborns and infants worldwide.

## Figures and Tables

**Figure 1 vaccines-12-00549-f001:**
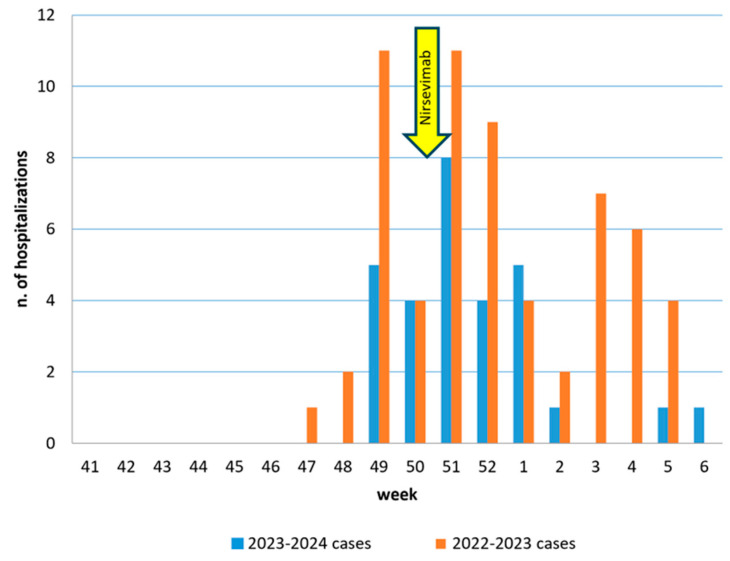
Hospitalizations for RSV bronchiolitis in Valle d’Aosta in the last two epidemic seasons.

**Figure 2 vaccines-12-00549-f002:**
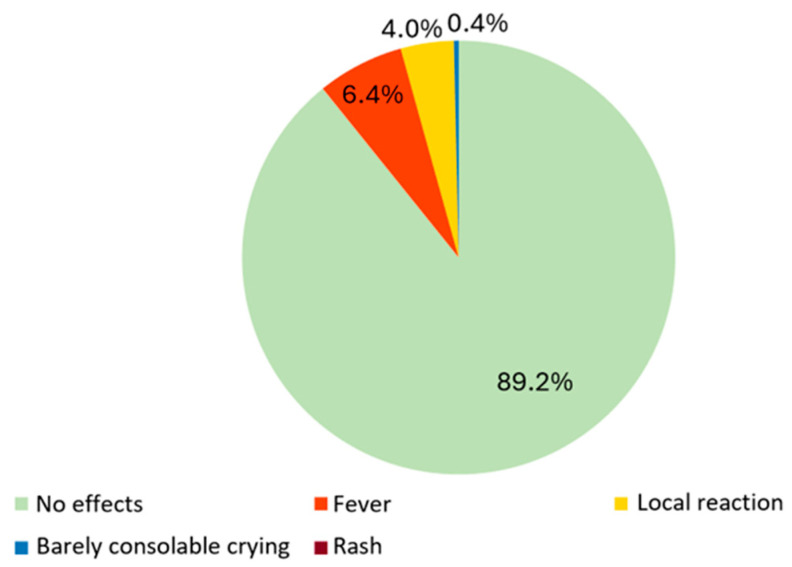
Side effects reported in the two weeks following nirsevimab administration. The percentage of rash was 0%.

**Table 1 vaccines-12-00549-t001:** Adherence to nirsevimab prophylaxis in those born in Valle d’Aosta, Italy from 1 May 2023 to 15 February 2024.

Period	Total Births	Candidates for Palivizumab	Non- Residents	Candidates for Nirsevimab	Treated with Nirsevimab	No Prophylaxis	Coverage
01/05/2023 18/12/2023	461	13	0	448	292	156	65%
19/12/2023 15/02/2024	95	3	3	89	77	12	86%
**Total**	556	16	3	537	369	168	69%

## Data Availability

All the data are contained within the article.

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
