# Peer review of "Safety and Efficacy of Nirsevimab in a Universal Prevention Program of Respiratory Syncytial Virus Bronchiolitis in Newborns and Infants in the First Year of Life in the Valle d’Aosta Region, Italy, in the 2023–2024 Epidemic Season"

_vaccines, 2024, doi:10.3390/vaccines12050549_

Round 1
Reviewer 1 Report
Comments and Suggestions for Authors
I appreciate the invitation to critically review the manuscript with ID vaccines-2969667, in which the researchers seek to describe the safety and efficacy profile of a monoclonal antibody to prevent RSV infections in a high-risk population. This addresses a topic of paramount importance; however, I have a series of comments that I would like to kindly bring to the attention of the research group.
Major comments
Comment 1: The study design is not specified anywhere in the manuscript. The wording is somewhat confusing and it is difficult to understand the methodology that was used. I suggest improving this aspect.
Comment 2: The authors do not present a comparison of the characteristics of the patients between the groups of interest, so the results presented, unless it is ensured that the groups were comparable in terms of the frequency of comorbidities and demographic aspects, lack validity.
Minor comments
Comment 3: I understand that it was not the aim of the study, but information on the average cost of treatment with the monoclonal antibody, and compared to standard therapy, would be very useful for readers.
Comment 4: There are aspects cited in the Conclusions section that seem more like prospects for study (i.e., 349-355), so they should be included in the Discussion section.
Author Response
I appreciate the invitation to critically review the manuscript with ID vaccines-2969667, in which the researchers seek to describe the safety and efficacy profile of a monoclonal antibody to prevent RSV infections in a high-risk population. This addresses a topic of paramount importance; however, I have a series of comments that I would like to kindly bring to the attention of the research group.
Re: Thank you very much for your suggestions. We revised the manuscript according to the recommendations.
Major comments
Comment 1: The study design is not specified anywhere in the manuscript. The wording is somewhat confusing and it is difficult to understand the methodology that was used. I suggest improving this aspect.
Re: Details have been added according to your recommendations (p. 3).
Comment 2: The authors do not present a comparison of the characteristics of the patients between the groups of interest, so the results presented, unless it is ensured that the groups were comparable in terms of the frequency of comorbidities and demographic aspects, lack validity.
Re: We added further details according to your suggestion (p. 5). We also clarified that subgroup analyses by stratifying the population on the basis of risk factors for serious RSV disease (such as prematurity, twins, gender, breastfeeding, parental smoking) could yield valuable insights into the differential effectiveness of nirsevimab across various risk strata (p. 9).
Minor comments
Comment 3: I understand that it was not the aim of the study, but information on the average cost of treatment with the monoclonal antibody, and compared to standard therapy, would be very useful for readers.
Re: The need of further studies on cost of treatment with nirsevimab has been highlighted in the Discussion (p. 7) and the Conclusions (pp. 7-8).
Comment 4: There are aspects cited in the Conclusions section that seem more like prospects for study (i.e., 349-355), so they should be included in the Discussion section.
Re: Done (pp. 8-9).
Reviewer 2 Report
Comments and Suggestions for Authors
Dear authors: congratulation to your paper
Safety and Efficacy of Nirsevimab in a Universal Prevention 2 Program of Respiratory Syncytial Virus Bronchiolitis in New- borns and Infants in the First Year of Life in the Valle d'Aosta Region, Italy, in the 2023-2024 Epidemic Season
Content:
This study describes the effectiveness 22 and safety of the universal prevention program of RSV bronchiolitis using the monoclonal antibody 23 nirsevimab, in children resident in Valle d'Aosta born during the 2023-2024 epidemic season
This study describes the effectiveness and safety of the universal prevention program of RSV bronchiolitis using the monoclonal antibody nirsevimab, in children resident in Valle d'Aosta born during the 2023-2024 epidemic season.
Well structured article
authors explain their paper in many contexts.
Authors are experts who understand this problem: for example:
Concrete description: In the Valle d'Aosta Region, like in other parts of Italy, recent epidemic seasons have 106 witnessed a notable surge in cases of RSV infections, coupled with an escalation in their 107 severity.
next lines 106-121...
excellent explanations - tabs. 1-2
Very important comment: :
Moreover, from an economic sustainability standpoint, the impact assessment must 259 encompass direct savings realized through reduced hospital admissions, cessation of se-260 lective prophylaxis with palivizumab, and indirect savings stemming from diminished 261 social care costs borne by parents for children afflicted by RSV infection (e.g., medical 262 expenses, lost work days) or enduring long-term consequences of RSV infection (e.g., re-263 current wheezing, asthma) [35]. These economic considerations further underscore the 264 value proposition of universal prophylaxis with nirsevimab in mitigating the burden of 265 RSV-related illnesses. 266
Interestingly, our findings
Recommendation - moreover, economic evaluations could assess the overall cost-effectiveness of 343 implementing universal nirsevimab prophylaxis compared to alternative preventive strat-344 egies, considering factors such as healthcare resource utilization and societal costs
Article is suitable for publishing.
Author Response
Dear authors: congratulation to your paper
Safety and Efficacy of Nirsevimab in a Universal Prevention 2 Program of Respiratory Syncytial Virus Bronchiolitis in New- borns and Infants in the First Year of Life in the Valle d'Aosta Region, Italy, in the 2023-2024 Epidemic Season.
Content:
This study describes the effectiveness 22 and safety of the universal prevention program of RSV bronchiolitis using the monoclonal antibody 23 nirsevimab, in children resident in Valle d'Aosta born during the 2023-2024 epidemic season
This study describes the effectiveness and safety of the universal prevention program of RSV bronchiolitis using the monoclonal antibody nirsevimab, in children resident in Valle d'Aosta born during the 2023-2024 epidemic season.
Well structured article
authors explain their paper in many contexts.
Authors are experts who understand this problem: for example:
Concrete description: In the Valle d'Aosta Region, like in other parts of Italy, recent epidemic seasons have 106 witnessed a notable surge in cases of RSV infections, coupled with an escalation in their 107 severity.
next lines 106-121...
excellent explanations - tabs. 1-2
Very important comment:
Moreover, from an economic sustainability standpoint, the impact assessment must 259 encompass direct savings realized through reduced hospital admissions, cessation of se-260 lective prophylaxis with palivizumab, and indirect savings stemming from diminished 261 social care costs borne by parents for children afflicted by RSV infection (e.g., medical 262 expenses, lost work days) or enduring long-term consequences of RSV infection (e.g., re-263 current wheezing, asthma) [35]. These economic considerations further underscore the 264 value proposition of universal prophylaxis with nirsevimab in mitigating the burden of 265 RSV-related illnesses. 266
Interestingly, our findings
Recommendation - moreover, economic evaluations could assess the overall cost-effectiveness of 343 implementing universal nirsevimab prophylaxis compared to alternative preventive strat-344 egies, considering factors such as healthcare resource utilization and societal costs
Article is suitable for publishing.
Re: Thank you very much for your positive evaluation and the acceptance of our manuscript. We revised the manuscript according to suggestions received from the other two reviewers.
Reviewer 3 Report
Comments and Suggestions for Authors
Thank you for asking me to review this interesting manuscript. This study described the effectiveness and safety of the universal prevention program of RSV bronchiolitis using the monoclonal antibody nirsevimab. The method and the results are clearly described. I believe that the nirsevimab has very good effectiveness and it’s safe. While I still have some comments which may help to make the conclusions more convince.
This is post-license study, the enrollment of study subjects was clearly described. But there is no information about the epidemic of RSV in 2023-2024 season at the same region. Although it may similar to the 2022-2023 season, we don't know it. The conclusion would be more convincing if the author can provide evidence of the RSV epidemic was similar. Then the decrease of the bronchiolitis hospitalization can conclude that is due to the effect of the universal antibody program but not the lower epidemic. Actually, when look at the results of two seasons, it seems the epidemic level of RSV in 2023-2024 (before the immunization) was lower than that in 2022-2023. Can the author provide some surveillance results for RSV in 2023-2024 from other age group in the same region or neighboring area? Which is acceptable if they did not have strong comparison group data.
Author Response
Thank you for asking me to review this interesting manuscript. This study described the effectiveness and safety of the universal prevention program of RSV bronchiolitis using the monoclonal antibody nirsevimab. The method and the results are clearly described. I believe that the nirsevimab has very good effectiveness and it’s safe. While I still have some comments which may help to make the conclusions more convince.
Re: Thank you very much for your comments. We revised the manuscript according to your suggestions and those received from reviewer #1.
This is post-license study, the enrollment of study subjects was clearly described. But there is no information about the epidemic of RSV in 2023-2024 season at the same region. Although it may similar to the 2022-2023 season, we don't know it. The conclusion would be more convincing if the author can provide evidence of the RSV epidemic was similar. Then the decrease of the bronchiolitis hospitalization can conclude that is due to the effect of the universal antibody program but not the lower epidemic. Actually, when look at the results of two seasons, it seems the epidemic level of RSV in 2023-2024 (before the immunization) was lower than that in 2022-2023. Can the author provide some surveillance results for RSV in 2023-2024 from other age group in the same region or neighboring area? Which is acceptable if they did not have strong comparison group data.
Re: Thank you for your suggestion. We added data on this issue (p. 7) with a specific reference (p. 12) supporting the effect of universal nirsevimab prophylaxis.